# Factors Associated with the Suspected Delay in the Language Development of Early Childhood in Southern Thailand

**DOI:** 10.3390/children9050662

**Published:** 2022-05-04

**Authors:** Namfon Rithipukdee, Kiatkamjorn Kusol

**Affiliations:** Department of Children and Adolescent Nursing, School of Nursing, Walailak University, Thasala, Nakhon Si Thammarat 80160, Thailand; namfon.pukdee@gmail.com

**Keywords:** factors, suspected delay in language development, early childhood

## Abstract

Many children have a suspected delay in language development and need extensive support from parents and the health care team. This study aimed to investigate the suspected delay in language development and the factors associated with the suspected delay in language development among early childhood in Southern Thailand. Children aged 24 to 60 months were recruited as study samples using stratified random sampling conducted in 23 districts and simple random sampling in 7 sections (425 children). The research instruments comprised demographic data on the children and families, the preschool temperament questionnaire, and the Language Development Screening questionnaire using the Developmental Surveillance and Promotion Manual and data collected from July 2020 to January 2021. Data analysis used descriptive statistics and logistic regression. The results showed that the suspected delay in development was 40.9 percent. Daily screen time exceeding 2 h per day (AOR = 17.30, 95% CI: 7.35–40.72), and regarding a child’s temperament, moderate-to-difficult temperament (AOR = 9.56, 95% CI: 5.12–17.85) was significantly associated with the suspected delay in language development. Genders, gestational age of early childhood, and the age of the caregivers were not associated with the suspected delay in language development. The study suggested that a decrease in daily screen time and appropriate responses will help develop language in children.

## 1. Introduction

Language capacity is an integral part of the life of children. Language development is a hierarchical change beginning with hearing and learning by hearing through the sound of words [1]. According to Piaget’s theory, before a child can speak, the child communicates through action and behaviors. After that, children begin to perform concrete thinking operations and develop expressive and receptive language [2]. After six months, children’s ability to distinguish sounds will have improved, followed by development concerning lexical resources and fast word interconnection from 1.5 to 3 years [3]. These skills significantly contribute to the development in early childhood in the long run. The assessment of early childhood development screening tools includes the Modified Denver, Denver Developmental Screening Test (DDST), Diagnostic Inventory for Screening Children (DISC), Developmental Skills Instrument (DSI), and Modified Denver II [4,5]. The screening of early childhood development in Thailand is mainly performed using the Developmental Surveillance and Promotion Manual (DSPM). DSPM in language development screening can be categorized into two aspects: receptive language comprehension through sensory nerves obtained through input from hearing and input from sight. With this, a child can differentiate differences in sounds, interpret them and grasp their meanings. Turning to expressive language is a verbal communication of intentions. The World Health Organization (WHO) disclosed that 8 percent of children under 5 years of age exhibit non-age-appropriate development [6]. In Thailand, data on the overall development of children in early childhood in 2017 suggested that 23 percent of the children in early childhood exhibited a suspected delay in development. More specifically, 39.87 percent of the children were potentially had postponed development, and 26.63 were imagined to have a delay in linguistic development in Health Area 11, Southern Thailand [7], which was higher than the overall national proportion.

Language development reflects how a child’s brain and nervous system function continuously through each inherent developmental stage [8,9]. Studies on a child’s linguistic progress have demonstrated that preterm infants undergo relatively more delayed development during the early years, especially in the domain of prescriptive language use than those born in full pregnancy term [10,11]. Preterm children aged 18 to 36 months were found to have more linguistic limitations than those born at full-term. The preterm children, for example, acquired vocabularies of less than 50 and had difficulty piecing together their vocabularies. The preterm children had a delay in linguistic development, some of whom experienced a delay in both using and understanding the language [12]. Additionally, studies regarding the relationship between gender and linguistic abilities suggested that female children demonstrated faster linguistic development than male children [13,14].

Temperament is an innate, intrinsic model unique to each child with outstanding resistance to change [15]. Children with easy temperaments can be understood as possessing effective emotional control leading to positive social interactions [16,17]. Delayed linguistic ability undermines a child’s temperament and personality [18]. On the other hand, those with difficult temperaments lessen others’ interest in talking to them, consequently compromising the child’s expressive language development and susceptibility to anxiety when exposed to unacquainted surroundings. Hence, changing from a familiar home to a school can become a catalyst for mutism in children [19,20].

Caregivers or parents play a significant role in enhancing a child’s linguistic ability in early childhood because they provide understanding, care, connection, and interaction with the child. Previous studies have shown that the predictive factors of age-appropriate development in early childhood include the mother’s age: chances for mothers aged 20–35 years and above 35 years to give birth to age-appropriate development in children decreased by 18 and 33 percent, respectively, compared to mothers less than 20 years. This is because such a group of mothers is a working-age population, meaning they have less time to care for their children. As a result, their children are prone to a suspected delay in development [21]. The studies conducted with preterm infants in low-income families have found that mothers’ age attributed to a delay in development [22].

On the contrary, Phongphetdit and Authawee have found that mothers aged 20 to 35 could raise children to the age-appropriate development stages because, as part of the working-age population, they were able to seek knowledge and access a wide variety of media easily and instantly [23]. Nevertheless, under some circumstances, parents’ or caregivers’ limited abilities and skills complicated the orientation of the home environment to cater to a child’s development, not to mention today’s technological dynamics evolving by leaps and bounds. Children in early childhood have become more attached to screen gadgets such as smartphones, tablets, computers, or even televisions [24]. The American Academy of Pediatrics pointed out that children aged 2 to 5 years could spend one hour a day watching quality media [25]. The careers of parents usually contribute to children watching television or surfing the internet alone for entertainment, or sometimes screen time is used to appease a child’s tantrums. These all result in a child’s delayed linguistic abilities [26]. This also contributes to sleep problems, impaired executive function and general cognition, and the relationship between parent and child in early childhood [27,28,29,30]. In addition, children with language delays or communication difficulties may be at an increased risk of learning disabilities and illiteracy, including reading and writing problems in adulthood [31,32]. The issues and importance of the factors mentioned above concerning a child’s development (including caregivers, children, environment, or changing social conditions) influence the child’s development; however, there is a lack of studies on language development in early childhood in Southern Thailand. Researchers have acknowledged the importance of investigating the situations related to language development and the factors associated with a delay in language development in early childhood to provide empirical data that can be applied for language development so that children in early childhood can reach language developmental milestones.

## 2. Method

### 2.1. Participants

This study is a descriptive study. The researchers collected data from the well-baby clinics in the Tambon Health Promoting Hospitals from July 2020 to January 2021. The research population comprised children aged 24 to 60 months and parents or caregivers residing in Southern Thailand. The researchers conducted randomized sampling in Nakhon Si Thammarat Province, which has a total children population of 92,973 persons. The researchers calculated the sample size using the Krejcie and Morgan formula, generating 382 research samples [33]. The researchers added about 10% to prevent the collection of incomprehensive data. Stratified random sampling was employed in 23 districts, followed by simple random sampling in 7 selected districts. Then, two of the Tambon Health Promoting hospitals were selected, yielding a group of 425 research samples. The inclusion criteria specified conditions for the caregivers and the children as follows: Caregivers may be parents or primary caregivers, anyone living with the child, and those providing regular care for longer than six months. The caregivers must also be 18 years of age or above, be of Thai ethnicity and Thai nationality, and communicate and understand Thai. The children must be within 24–60 months, visit the well-baby clinic for vaccination, and be free from genetic disorders and any other disease that impacts child development such as down syndrome, mental retardation, autistic disorder, etc. The children must exhibit normal visual, and learning abilities.

### 2.2. Measures

The research instruments are as follows:The demographic data questionnaire for the children, parents or caregivers, and families (14 items):1.1Demographic data of the children, including genders, religions, age, gestational age, delivery methods, birth weight, the average number of hours spent watching television, videos, smartphones, tablets, telephone, and computer games per day, and child temperament;1.2Demographic data of the parents or caregivers and families, including age, relationship with the children, family characteristics, levels of education, occupations, and monthly household income.The preschool temperament questionnaire: The researchers employed the research instrument developed by Nattaya Sangsai et al. (2011), with a reliability value of 0.80 and content validity of 0.83 [34]. The questionnaire consists of 36 items featuring characteristics of 6 aspects of temperament: Activity level (8 items), Rhythmicity or Regularity (5 items), Approach/ withdrawal (5 items), Adaptability (5 items), Intensity of Reaction (7 items), and Mood (6 items). Questions were answered using a 5-point rating scale from 36 to 180 points. Interpretations of the children’s temperament were clustered into two groups: those who scored 36–132 were categorized as children with moderate-to-difficult temperament, and those who scored 133–180 were classified as children with an easy temperament.The language development screening questionnaire: The researcher screened a child’s development with the Developmental Surveillance and Promotion Manual (DSPM), divided into Receptive language and Expressive language developed by Siriporn Kanchana et al. [35]. The questionnaire yielded a sensitivity value of 96.04 and a specificity value of 64.67. The interpretation was categorized as either age-appropriate development (1 score) or suspected delay (0 scores). As demonstrated by Cronbach’s alpha, the questionnaire’s reliability was 0.81.

### 2.3. Statistical Analyses

Data were analyzed using the SPSS^®^ Version 24.0 for Windows™ (IBM Corporation, New York, NY, USA). The number of suspected delays in language development and demographic data was analyzed using frequency distribution, percentage, mean, and standard deviation (S.D.) The factors’ correlation with suspected delay in language development was assessed using binary logistic regression.

## 3. Results

Regarding the children’s language development in early childhood, 59.1 percent exhibited age-appropriate development, while 40.9 percent showed a suspected delay in development, as shown in Table 1.

### 3.1. Demographic Data of Children

Most of the children were Buddhists (72%), and their average age was 40 months. In addition, 397 of the children (93.41%) were full-term infants, and 76% were born through vaginal delivery. Forty-eight percent of children’s birth weights fell into a range of 2500 to 3000 g. More than half of the children in early childhood (73.6%) were reported to have less than 2 h of average daily screen time (i.e., television, videos, smartphones, tablets, or computer games). The children demonstrated easy temperament, as the children’s average temperament score was 105.99, as displayed in Table 2.

### 3.2. Demographic Data of Parents or Caregivers

Most caregivers were from 25 to 35 years of age (58.60%), and more than half of the caregivers were mothers (80.24%). In addition, 62.82% were single-family, 53.41% obtained a secondary school degree, and almost one-third (28%) were unemployed. Finally, more than half (53.64%) earned 10,000–20,000 Thai baht per month, as displayed in Table 3.

The binary logistic regression analysis suggested that the children with more than 2 h of daily screen time were significantly associated with a suspected delay in language development (AOR = 17.30 (95% CI: 7.35–40.72). In terms of temperament, the children exhibiting moderate-to-difficult temperament were significantly associated with a suspected delay in language development (AOR = 9.56; 95% CI: 5.12–17.85). Finally, gender, gestational age, and caregivers’ age groups were not associated with a suspected delay in language development, as displayed in Table 4.

## 4. Discussion

The analysis results yielded in the research sample group of average 40 months-old children ($\bar{x}$ = 40.90, S.D. = 9.04) indicated that they were mainly delivered at full-term and had an age-appropriate birth weight. For a mother assuming the role of the primary caregiver in a single-family home and having obtained a secondary school degree, the data suggested that in terms of the language development situation of Southern Thai children in early childhood, nearly half exhibited a suspected delay in language development (40.90%). The analysis further illuminated that the children that spent more than 2 h on daily screen time were significantly associated with a suspected delay in language development. Furthermore, the children categorized as having moderate-to-difficult temperament were significantly associated with a suspected delay in language development compared to those with easy temperament. At the same time, gender, gestational age, and caregivers’ age groups were not associated with a suspected language development delay.

This study suggests that children with more than 2 h of daily screen time were associated with a suspected delay in language development. This was consistent with the American Academy of Pediatrics and World Health Organization that advised against screen time for children below two years of age or a daily maximum 1 h limit; in addition, quality programs should be allowed [26,36,37]. This is due to the fast-paced technological evolvement, which has made technology more accessible for children, resulting in the children’s passive interaction in a non-virtual domain. In addition, a prolific increase in young children’s screen time may derive from multiple causes, one of which is the essential caregivers and the home environment. If, for instance, a mother permits a child screen time even in the mother’s presence during meals as a common stratagem to stop the child’s naughty behavior or to manipulate the child’s behavior rather than doing other activities, an increase in screen time will subsequently follow [38,39]. In the context of a traditional Thai family, some children live in an extended family together with their grandparents, who perceive the use of screen time as entertainment for their grandchildren and, as a result, allow them to indulge their cravings for screen time without experience in operating these gadgets themselves [40]. This study, in most cases, found that caregivers or parents filled their children’s time with the use of gadgets, the simplest case of which was all-day television watching while they were away for a full-time job schedule or tending to house chores The reason was rooted in the parents’ need to assume other responsibilities besides caring for their child. Unaccompanied children might be drawn to programs featuring age-inappropriate content [41]. Exposure to excessive screen time (>6.5 h/day) via television, mobile phones, iPads/tablets, or computers is likely to inhibit verbal interaction or communication between the child and the caregiver, resulting in even greater susceptibility to a delay in language development [42,43]. Therefore, over the age period from 0 to5 years, a child’s learning should originate from outdoor activities, physical contact, or storytelling because the input received through the sensory channels involved relatively accelerates a child’s learning progress in the acquisition, problem-solving, and enthusiasm to learn more than the screen can deliver [26,40].

In addition, this study suggested that the children with moderate-to-difficult temperaments were associated with a suspected delay in language development compared to those with easy temperaments. This may be related to the fact that character in early childhood is mainly governed by genetic and environmental predispositions even though it becomes gradually nurtured with older age. A child’s character is indicative of how they respond to the environment; for example, those with moderate-to-difficult temperament may cause caregivers to provide inappropriate care, followed by an even worse temperament exhibited by the child [44]. Difficult temperament impedes parenting and caregiving in almost every sphere, including eating, sleeping, excreting, and temper control. There is a possibility that this may be detrimental to a child’s ability to learn and control emotions [45], which decreases people’s desire to converse with the child, potentially hindering their linguistic capacity. Over the age period from 2 to 5 years, children naturally begin to make evident linguistic progress in both receptive and progressive language to convey meanings, express opinions, and live with others through reception, interpretation, decision-making, and expression or gestures. Equipped with an easy temperament, a child can maintain a good mood, easily get along with others, acquire novel information, and effectively communicate with peers, solidifying the child’s linguistic ability to a greater extent. In addition, caregivers can offer stimuli, assistance, and caregiving to promote children with problematic temperaments in linguistic development to communicate better [46]. Caregivers’ strategies that emphasize a family’s relationship, prompt attention to a child’s needs, closeness, and attachment also positively affect the child’s language development [47].

On the contrary, other factors, including gender, gestational age, and caregivers’ age groups, were not associated with a suspected language development delay. Of the 174 children in this study that had a suspected language development delay, 37.80% were females and 45.10% were males. Research reports have suggested that male children were more susceptible to a suspected delay in language development than their female counterparts. This is consistent with the study on the correlation between anthropometric indices at birth and a developmental delay in children aged 4–60 months in Isfahan, where there was an increased rate of suspected delay in language development among male children [48]. Despite the lack of tangible evidence, the climbing tendency of the suspected delay in language development in male children is greater than that in female children, since abnormalities of the X-link are common among males. These are involved in a child’s suspected delay in language development [49,50]. In this study, 90 percent of the research samples were born at full-term. Therefore, gestational age did not have a dominant effect on the suspected delay in language development. However, some studies have shown that infants born prematurely, weighing between 1500 to 1999 g and delivered using a cesarean delivery, were prone to delayed development [51,52,53]. Most caregivers were within a 25–35-year age range and belonged to the working population, so they were more prepared for proper child-raising. Chances for mothers above 35 years to have a child with a suspected delay in development are higher than for those 25–35 years old. However, children born to young mothers have an increased rate of exhibiting a suspected delay in development in all aspects at 60 months. Such a risk can be reduced when children are born to older-aged mothers, and the correlation between health conditions and better development of children in early childhood is examined [54,55].

## 5. Conclusions

Delay in language development may include difficulties in learning and short attention spans, depriving the child of learning abilities at successive advanced levels. The study results revealed that the factors associated with a suspected delay in language development in early childhood include daily screen time exceeding 2 h and a moderate-to-difficult temperament. Therefore, nurses and health care providers should be aware of these factors and give information to caregivers to reduce the factors contributing to daily screen time and to provide appropriate responses that will help in a child’s language development.

## 6. Limitations and Scope of Future Research

Even though this study was exhaustive, some limitations still exist as follows. There were limitations regarding the location and sample selection. This study involved 425 children in early childhood who resided in Nakhon Si Thammarat Province. Nakhon Si Thammarat is one of the provinces in Southern Thailand and is considered an urban area, but various fields of employment for parents and caregivers were not taken into consideration in the rural areas. Thus, future studies should include children in early childhood and caregivers from all provinces in Thailand, including urban areas, rural areas, and parents or caregivers from various fields of employment. Therefore, the findings can be generalized to the early childhood population in Thailand.

## Figures and Tables

**Table 1 children-09-00662-t001:** The children’s language development status (*n* = 425).

Language Development	Number	%
Children with normal language development	251	59.10
Children with a suspected delay	174	40.90

**Table 2 children-09-00662-t002:** The demographic data of children (*n* = 425).

Demographic Data of Children	Number	%
Gender		
Male	184	43.29
Female	241	56.71
Religion		
Buddhism	306	72.00
Islam	119	28.00
Age groups (Range 24–60, $\bar{x}$ = 40.90, S.D. = 9.04)		
24–36 months	152	35.76
37–48 months	183	43.06
49–60 months	90	21.18
Gestational age (weeks)		
<37	28	6.59
≥37	397	93.41
Type of delivery		
Normal	323	76.00
Cesarean	102	24.00
Birth weight (g)		
<2500	71	16.71
2500–3000	204	48.00
>3000	150	35.29
Daily screen time (hours)		
<2	313	73.60
>2	112	26.40
Child’s temperament (Score range 59–169, $\bar{x}$ = 105.99, S.D. = 29.84)		
Easy temperament	202	47.50
Moderate to difficult temperament	223	52.50

**Table 3 children-09-00662-t003:** The demographic data of parents or caregivers (*n* = 425).

Demographic Data of Parents or Caregivers	Number	%
Age (years)		
<25	71	16.70
25–35	249	58.60
>35	105	24.70
Family characteristic		
Single-family	267	62.82
Extended family	158	37.18
Relationship with the children		
Father	34	8.00
Mother	341	80.24
Grandmother/Grandfather	44	10.35
Other	6	1.41
Levels of education		
Primary education	58	13.65
Secondary education	227	53.41
Diploma education	56	13.18
Undergraduate/Graduate degree	84	19.76
Occupation		
Agriculturist	82	19.29
Government employee/State Enterprises	28	6.59
Employment	23	5.41
Private business	66	15.53
Trader	107	25.18
Unemployed	119	28.00
Household income/ month (Thai baht)		
<10,000	138	32.47
10,000–20,000	228	53.64
20,001–30,000	52	12.24
>30,000	7	1.65

**Table 4 children-09-00662-t004:** Binary logistic regression analysis for exploring factors associated with a suspected delay in language development (*n* = 425).

Variable		Language Development Status		B	SE	Wald	*p*-Value	Crude OR (95% CI)	Adjusted OR (95% CI)
		Normal N (%)	Suspected N (%)						
Gender	Female	150 (62.20)	91 (37.80)					1	1
	Male	101 (54.90)	83 (45.10)	0.28	0.28	0.96	0.322	1.35 (0.91–2.00)	1.32 (0.75–2.33)
Gestational age (weeks)	≥37	237 (59.70)	160 (40.30)					1	1
	<37	14 (50.00)	14 (50.00)	0.28	0.56	0.25	0.61	1.48 (0.68–3.19)	1.33 (0.44–4.02)
Daily screen time (hours)	<2	244 (78.00)	69 (22.00)					1	1
	>2	7(6.25)	105 (93.75)	2.85	0.43	42.60	0.00	53.04 (23.58–119.28)	17.30 (7.35–40.72)
Age of caregivers (years)	<25	38 (53.52)	33 (46.48)					1	1
	25–35	152 (61.04)	97 (38.96)	−0.22	0.39	0.32	0.57	1.20 (0.65–2.20)	0.79 (0.36–1.73)
	>35	61 (58.09)	44 (41.91)	0.10	0.44	0.05	0.81	0.88 (0.55–1.40)	1.10 (0.46–2.62)
Child’s temperament	Easy temperament	185 (91.58)	17 (8.42)					1	1
	Moderate-to-difficult temperament	66 (29.60)	157 (70.40)	2.25	0.31	50.28	0.00	25.88 (14.58–45.95)	9.56 (5.12–17.85)

## Data Availability

The data supporting this study’s findings are available from the corresponding author upon reasonable request.

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
