# Peer review of "Factors Associated with the Suspected Delay in the Language Development of Early Childhood in Southern Thailand"

_children, 2022, doi:10.3390/children9050662_

Round 1

Reviewer 1 Report

The authors have presented a potentially interesting paper about the risk factors associated with language delay in Thailand.

The manuscript seems to represent a novel contribution to studies in Thailand. However, I had difficulties following the authors text and meaning and there are some major aspects that should be tackled before resubmitting this paper.

What follows are my best guesses about issues that should be addressed in the manuscript.

Introduction

  • The theoretical framework is briefly alluded and not coherent; no clarifying information on the specific theoretical framework is provided.

  • Why this study should be conducted? What does this study bring to the discussion of language delay that every other study cited does not?

  • An expansion of the literature would increase the value of the work. I suggest to include more articles from high impact journals.

  • The definition of language is imprecise. While I agree with the distinction of the different components of language, it remains unclear what is the theoretical basis for the two grouping.

Participants

Statements regarding children IQ, visual and hearing impairments and other diagnoses or neurological disorders are needed.

Data collection tool

I suggest to replace the title of this section with “Measures”.

Results

  • Title of Tables are not clear.

  • In Table 1: “Children with normal”, what do the authors mean?

  • Line 153 and line 162: I suggest to change “percent” with %

  • Line 155“virginal delivery”: I think the authors mean “vaginal delivery”

  • The results of the Logistic Regression are not correctly reported both in the manuscript and in the table: B, SE, Wald and p value are missing.

Discussion and conclusion

  • The discussion, reads a little vague, it cannot be interpreted satisfactorily and lacks of coherence. I suggest to add more references and to discuss the results with respect to the existing evidence.

  • The conclusion includes unsupported claims; i.e., “Therefore, nurses, interdisciplinary teams, caregivers, and others involved should be aware of these factors and give them to reduce the factors and promote a child's age-appropriate development by two-way communication and appropriate responses will help develop children's language.” The two-way communication was never cited before. Moreover, since no intervention with children was conducted, arguing that appropriate responses will promote children’s language seems inappropriate. A discussion of possible further research paths would be more useful.

  • A “Limitations” section must be added.

Author Response

Dear Reviewer of the Journal

Please find the attached file that contains our manuscript entitled "Factors associated with the delayed language development of early childhood in Southern Thailand" First of all, I am very gladful and thank you very much for the opinions and suggestions of experts to make my manuscript clearer, more appropriate and more accurate. And I have revised already the issues as the expert suggestion, and edit of the English language with the proof of experts who use English as their native language.

Response to Reviewer 1 Comments

Point 1: Introduction: The theoretical framework is brief; Why this study should be conducted?; What does this study bring to the discussion of language delay? An expansion of the literature.

Response 1: I have additional adjustments to the theoretical language development to the introduction in paragraph 1. I have additional adjustments to the reason brought to the discussion of language delay in the last paragraph. An expansion of the literature includes more articles in the introduction in each paragraph.

Point 2: Participants: Statement regarding children in this study

Response 2: I have additional adjustments to the characteristics of participants already

Point 3: Data collection tool suggest to replace with Measures

Response 3: I have additional adjustments already

Point 4: Results; suggestion in title of the table, the meaning in Children with normal, percent with %, and the result of logistic regression

Response 4: I have additional adjustments already in the title of the table and table 4, Children with normal = normal language development.

Point 5: Discussion, and conclusion, Limitations

Response 5: I have additional adjustments already

Thank you so much for being so attentive to our manuscript.

Sincerely,

Kiatkamjorn Kusol

Reviewer 2 Report

This topic is important to our current climate. The references cited differ from what the abstract of the paper reports. This may be because of interpreting the paper by authors, but if so, please state it clearly. The introduction may require further elaborating on why this is an important topic. 

Consider limitations as an additional and strength of your paper. This will add value to this work.

Author Response

Dear Reviewer of the Journal

Please find the attached file that contains our manuscript entitled "Factors associated with the delayed language development of early childhood in Southern Thailand" First of all, I am very gladful and thank you very much for the opinions and suggestions of experts to make my manuscript clearer, more appropriate and more accurate. And I have revised already the issues as the expert suggestion, and edit of the English language with the proof of experts who use English as their native language.

Response to Reviewer 2 Comments

Point 1: Introduction: Why this study should be conducted?; An reference., abstract

Response 1: I have additional adjustments to the abstract, and the reason brought to the discussion of language delay in the last paragraph. A reference includes more articles in the introduction in each paragraph.

Point 2: A reference in the discussion

Response 2: I have additional adjustments already

Point 3: Limitations

Response 3: I have additional adjustments already

Thank you so much for being so attentive to our manuscript.

Sincerely,

Kiatkamjorn Kusol

Round 2

Reviewer 1 Report

The author has answered to all my previous criticisms.

Reviewer 2 Report

Thank you for addressing my comments. I appreciate authors addressing and elaborating the rationale to provide clarity to the audience.